# Adherence to the Mediterranean Diet and COVID-19: A Segmentation Analysis of Italian and US Consumers

Francesca Gerini, Tommaso Fantechi, Caterina Contini, Leonardo Casini and Gabriele Scozzafava *

Department of Agriculture, Food, Environment and Forestry, University of Florence, P.le delle Cascine 18, 50144 Firenze, Italy; francesca.gerini@unifi.it (F.G.); tommaso.fantechi@unifi.it (T.F.); caterina.contini@unifi.it (C.C.); leonardo.casini@unifi.it (L.C.)
* Correspondence: gabriele.scozzafava@unifi.it

**Abstract:** The COVID-19 pandemic has led many countries to implement restrictions on individual freedom to stop the contagion. The imposition of lockdowns has affected many socio-economic aspects and, in particular, eating habits, highlighting the need to analyse the healthiness of new consumption patterns. The aim of our study was to investigate the changes in adherence to the Mediterranean diet, a dietary model universally recognized as healthy, that have occurred both during and since the lockdown. The subsequent profiling of consumers allowed us to understand which sociodemographic and psychographic factors favoured the development of more or less adherence to Mediterranean diet consumption patterns. The study was conducted by administering a questionnaire to a representative sample of Italians and New Yorkers. Both groups, defined by deep socio-economic differences and by their own eating habits compared to the Mediterranean diet model, were affected by similar lockdown measures. The data collected were processed by cluster analysis that allowed to identify four homogeneous groups with respect to the adherence to the Mediterranean diet model. The results highlight a worrying situation with respect to the impacts of the pandemic on maintaining a proper dietary style according to the principles of the Mediterranean diet. In fact, there has been a general worsening trend due to an increase in consumption, in part linked to emotional eating, which is a cause for concern about the potential future impacts on the health of consumers. The study highlights the need, therefore, to implement actions by public decision-makers aimed at raising the awareness of citizens on the issue of correct eating habits and at developing adequate food policies to stem the trend towards unhealthy diets.

**Keywords:** Mediterranean diet; coronavirus; food consumption; healthy eating; consumer segmentation

## 1. Introduction

During the national lockdowns around the world enforced to contain the COVID-19 pandemic, restrictions and confinement within homes had major impacts on society and the economy [1]. Changes that occurred during lockdown have affected people's mental and physical health and habits [2–4]. Major dynamics have also affected the way people purchase food, their consumption habits, and the composition of diets [5–7]. In particular, unhealthy nutritional behaviours appear to be emerging globally, giving rise to concern for the health of citizens [8–10]. The states of fear, boredom, stress, and anxiety caused by the pandemic and isolation have led to emotional eating, i.e., the overconsumption of foods rich in fat and sugar that can provide a sense of gratification [11–13]. Not being able to go often to the supermarket has led to an increase in the consumption of long-life products such as frozen and canned foods, and a decrease in the consumption of fresh, healthy produce, such as fruit and vegetables [8,14–16]. Bad eating behaviours are therefore the result of this situation, which, combined with reduced physical activity during the lockdown, can negatively affect consumer health in both the short and long term [9,17]. Indeed, a healthy diet and physical activity play a key role in preventing diseases, such as cancer, diabetes, obesity, and hypertension [18].

The Mediterranean diet (MD) is considered by the international scientific community as a healthy and balanced dietary pattern capable of preventing the onset of many diseases [19,20]. Since the 1990s, the MD model has been presented as a pyramid showing the proportions and consumption frequencies of servings of the main food groups [21]. The diet is based on consuming larger amounts of cereals, fruits, vegetables, and olive oil, and moderate amounts of dairy products, eggs, and fish, and low amounts of meat and sweets [22]. As regards the health benefits, scientific research has identified a direct correlation between individuals' adherence to the MD and reduced mortality due to cancer and cardiovascular diseases [23–25]. In addition, it is associated with a protective effect against chronic disorders, decreased atherosclerotic events [26], and a reduction in cases of obesity [27] and diabetes [28]. During the lockdown, medical associations recommended eating foods belonging to the MD as they may have a therapeutic role in conditions associated with COVID-19 infection [29]. In fact, researchers have observed a lower risk of COVID-19 associated with a higher adherence to the MD [30,31]. In this sense, MD's anti-inflammatory properties and nutritional profile rich in vitamins, mono-unsaturated fatty acids and flavonoids are capable of improving the immune system response and attenuating the severity of COVID-19 [32,33].

Several studies have investigated consumer adherence to the MD before the pandemic and during the lockdown, e.g., [11,34–38]. Their main objective was to determine whether lifestyle changes brought by the pandemic positively or negatively affected diet quality. Most of these studies assessed MD adherence by applying the Mediterranean Diet Adherence Screener (MEDAS) score. A large study conducted in 16 European countries showed higher MEDAS scores during the lockdown compared with the previous period in all the countries surveyed [9]. The highest increases in the MEDAS score were recorded in Greece and North Macedonia, while the highest scores during the lockdown were detected in Spain and Portugal. Improved MD adherence was also observed in Croatia, especially among female respondents, aged 20 to 50 years, with a high level of education and a normal body mass index [36]. A study conducted in Italy found a moderate adherence to the MD (score between 6 and 9) in the majority of respondents (73.5%), who were mainly young people aged 18 to 30 years [39]. It also reported an increase in frozen food consumption and a decrease in alcohol consumption. This indicates that, during the lockdown, although Italians consumed more MD foods, these could not be purchased fresh as recommended by the diet model because of the supply difficulties triggered by the pandemic. In Cyprus, MD adherence remained moderate although a large number of consumers achieved a high score (32%) [35]. In that country, the consumption of healthy foods such as fruit, vegetables, and legumes increased as did the consumption of junk food including sweet beverages and commercial sweets. Studies on dietary changes during the pandemic in the United States found that older people followed a healthier diet than younger people. However, no significant changes in diet were detected during the lockdown [40,41]. These results could be due to the different restrictions adopted to curb the pandemic across the United States. As a matter of fact, there was no nationwide lockdown [42].

Studies that used the MEDAS score to assess dietary quality during the lockdown found an overall improvement in diet in all countries, whereas studies using other methodologies had varied results. A recent review by González-Monroy et al. [43] analysed the findings of 23 studies and found that in several countries adherence to healthy diets had worsened as a result of increased consumption of snacks, sweets, ultra-processed foods, and alcohol and decreased consumption of fruit, vegetables, and fresh foods. Other studies have also found a worsening of diets during lockdown in countries such as Italy, France, and Spain [44–47]. It thus follows that the different methodologies used may lead to different assessments of the quality of the diet followed during the pandemic. In this sense, results obtained in a same country with different methodologies should be integrated and compared to describe more comprehensively how diet changed during the pandemic phases.

Recent literature has investigated in depth the change in eating habits that occurred during the lockdown, e.g., [11,35–37]. However, it is yet to be seen whether these trends continued even in the following months and whether consumers have returned to their pre-lockdown eating behaviours. An early study carried out in China observed that irrational and unhealthy behaviours engaged in during the lockdown continued in the post-lockdown period [48]. Research in Italy found that compared to the lockdown, in the post-lockdown period, Italians reduced their consumption of healthy foods such as fruit and vegetables [49].

In this context, our study aimed to analyse how adherence to the MD has changed following the pandemic in Italy and the United States (State of New York). These two contexts, profoundly different in socioeconomic terms and with respect to dietary habits [31], both experienced a similar form of lockdown to contain the COVID-19 pandemic. In Italy, the lockdown was enforced on 9 March 2020, and was in effect until 18 May, when businesses began to gradually open again [50]. In the State of New York, the shutdown period was enforced progressively from 16 March 2020, and re-openings began on 8 June [51].

Specifically, the study posed the following research questions:

RQ1: How did adherence to the Mediterranean diet change during the pandemic? Specifically, since the lockdown ended, have people returned to their eating behaviours from before the lockdown?

RQ2: Were there any differences between Italy and the United States?

RQ3: Who are the consumers who improved, maintained, or saw a worsening in their diet?

To answer the research questions, we conducted an online survey in Italy and the United States with the aim of collecting data on the consumption of major MD food categories before the pandemic, and during and after the lockdown enforced to contain the COVID-19 pandemic. The consumption trends identified could guide policymakers in adopting public policy measures aimed at promoting healthier eating habits.

## 2. Data and Methods

### 2.1. Survey and Sample

We conducted an online survey in Italy and the United States (New York State) in September 2020. This month was representative of a post-lockdown situation. In fact, despite the shutdowns after this date due to the re-exacerbation of the health crisis, in September 2020, the pandemic was under control in the two countries of our survey, in terms both of the number of infected people and deaths and the number of those hospitalised. The questionnaire was administered by an international marketing research company (Toluna, Inc., Wilton, CT, USA) to its online panel. The sample consisted of 1228 respondents, including 726 from Italy and 502 from the United States (Table 1), and was representative of gender and age.

**Table 1.** Sociodemographic characteristics of the sample.

| | Italy Absolute Figures | % | USA Absolute Figures | % | Total Sample Absolute Figures | % |
|---|---|---|---|---|---|---|
| *Gender* | | | | | | |
| Male | 351 | 48 (49) | 229 | 46 (49) | 580 | 47 |
| Female | 375 | 52 (51) | 273 | 54 (51) | 648 | 53 |
| *Age* | | | | | | |
| 18–34 | 161 | 22 (23) | 137 | 27 (31) | 298 | 24 |
| 35–54 | 291 | 40 (36) | 220 | 44 (33) | 511 | 40 |
| >55 | 274 | 38 (41) | 145 | 29 (36) | 419 | 36 |
| *Number of household members* | | | | | | |
| 1 | 68 | 9 | 100 | 20 | 168 | 14 |
| 2/3 | 412 | 57 | 248 | 49 | 660 | 54 |
| 4+ | 246 | 34 | 154 | 31 | 400 | 32 |

**Table 1.** *Cont.*

| | Italy Absolute Figures | % | USA Absolute Figures | % | Total Sample Absolute Figures | % |
|---|---|---|---|---|---|---|
| *Number of children under 18 years old in the household* | | | | | | |
| 0 | 68 | 9 | 100 | 20 | 168 | 14 |
| 1/2 | 558 | 77 | 277 | 55 | 835 | 68 |
| 3+ | 100 | 14 | 125 | 25 | 225 | 18 |
| *Total respondents* | 726 | | 502 | | 1228 | |

Note: Census data referring to Italy [52] and State of New York [53] are shown in brackets.

As regards respondents from the United States, a filter was applied to consider only those residing in the State of New York, since that was the only area where the lockdown enforced was similar to Italy's in terms of duration and type of restrictions [50,51].

The procedures performed in this study are in accordance with the ethical standards of the institutional and/or national research committee and with the 1975 Helsinki declaration and its later amendments or comparable ethical standards. Informed consent was obtained from all participants involved in the research.

### 2.2. The Questionnaire and Definition of the Index of Adherence to the Mediterranean Diet

The questionnaire administered online was divided into multiple sections. The first part was aimed at ascertaining the requirements for participation in the survey, i.e., being a resident of Italy or New York State and being responsible for grocery shopping in the household. The second section contained a series of questions designed to survey consumption of food categories before the pandemic, and during and after the lockdown. Following the approach proposed by Cavaliere et al. [54], we considered the different food items covering all the food groups of the MD pyramid [19,55] by collecting their weekly consumption frequency according to a 5-point Likert scale, where 1 was equal to "Never" and 5 to "Several times a day". Individual MD adherence was calculated according to the model used by Cavaliere et al. [54] by assigning a score from 0 to 2 for each food according to its frequency of consumption in accordance with the MD pyramid guidelines. By summing these scores, we obtained three indices of MD adherence for each respondent: adherence before the pandemic (ABP), adherence during lockdown (ADL), and adherence after lockdown (AAL). In addition, for each food category, respondents were asked whether consumption increased, decreased, or was unchanged during the lockdown compared with before the pandemic and in the post-lockdown period compared with the lockdown period.

The following section was devoted to collecting psychographic information from respondents using the scale of the single-item Food Choice Questionnaire developed by Onwezen et al. [56] regarding the importance of 11 attributes related to foods consumed daily. Respondents were asked to answer to the statement 'It is important to me that the food I eat on a typical day . . . ': is healthy, is a way of monitoring my mood, is convenient, provides me with pleasurable sensations, is natural, is affordable, helps me control my weight, is familiar, is environmentally friendly, is animal friendly, and is fairly traded. The answers were recorded on a 5-point Likert scale, where 1 was equal to "Strongly disagree" and 5 to "Strongly agree". Finally, the last section of the questionnaire concerned the collection of sociodemographic information of the survey participants.

### 2.3. Cluster Analysis and Profiling

In order to identify homogeneous segments of consumers defined by similar adherence to the MD before the pandemic, and during and after the lockdown, a cluster analysis was carried out using Stata 17 software [57]. Ward's method was used for clustering. It is based on a hierarchical clustering algorithm and a squared Euclidean distance [58]. The segmentation variables included in this analysis were ABP, ADL, AAL, and the variable considering the difference between adherence to the MD after lockdown and before the pan-

demic (DeltaAft_Bef). In order to determine the optimal number of clusters, the pseudo-F statistic, as defined by Caliński et al. [59] and the Duda et al. [60] Je(2)/Je(1) index, were calculated for 2- to 10-cluster solutions. According to both rules, larger indices indicate more distinct clustering. Stata presents the Duda/Hart index and a pseudo T-squared value. The rule of thumb for choosing the number of clusters to keep indicates that smaller pseudo T-squared values correspond to more distinct clustering [61]. The choice of the number of clusters was made by combined reading of the pseudo-F, Je(2)/Je(1), and pseudo T-squared indices.

The profile of the identified clusters was described by chi-squared automatic interaction detection (CHAID) analysis [62]. CHAID is an iterative technique that employs stepwise Chi-squared tests to determine which classification variables (predictors) are most effective in characterising clusters (dependent variable). Then, the population was divided based on significant categories (levels) of the predictor. The predictors that were included in the CHAID analysis to profile the clusters were sociodemographics, psychographics, and food consumption. Figure 1 shows the graphical representation of the study's implementation.

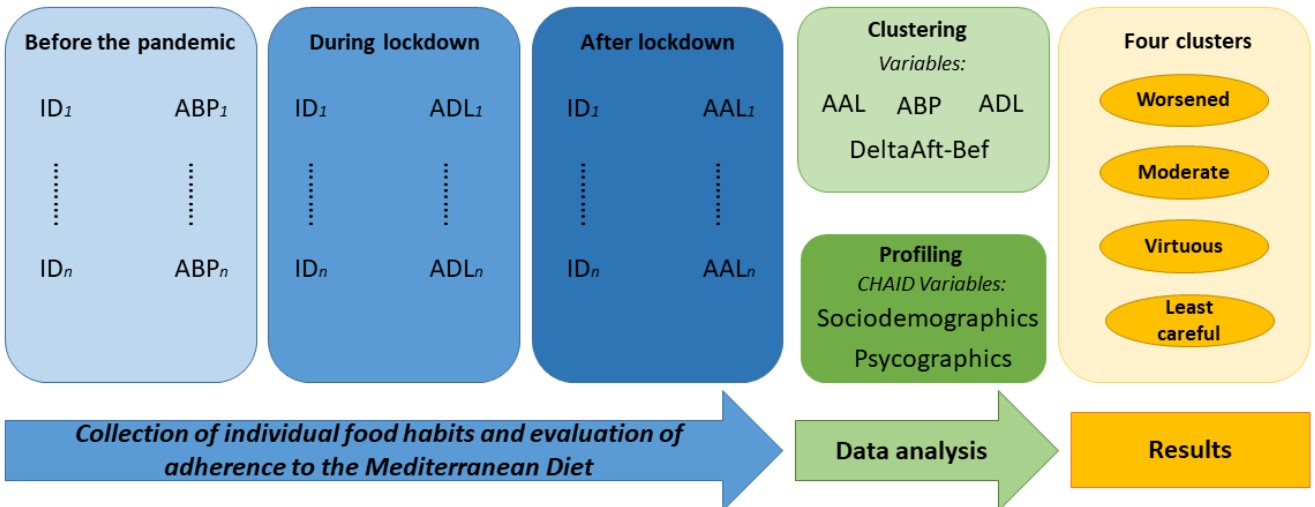

**Figure 1.** Visual representation of the study. (Notes: For each respondent (ID$_i$), ABP is adherence to the MD before the pandemic, ADL is adherence to the MD during lockdown, AAL is adherence to the MD after lockdown and DeltaAft-Bef is the difference between AAL and ABP).

## 3. Results

### 3.1. Clusters Identified Based on Adherence to the Mediterranean Diet

The total sample, consisting of 1228 Italian and US consumers, showed a mean ABP of 16.17, ADL of 13.48, and AAL of 13.27. The Italian sub-sample showed a higher mean adherence to the MD than the US sub-sample over the three periods observed. Specifically, Italians reported an ABP index of 17.81 (compared with 13.80 for the US sample), an ADL of 14.72 (compared with 11.69 for the US sample), and an AAL of 14.61 (compared with 11.33 for US sample). Therefore, the index trends show that during the pandemic, consumers progressively decreased their adherence to the MD.

The cluster analysis was implemented by applying the criteria suggested by Caliński et al. [59] and Duda et al. [60]. Considering both rules and the dendrogram analysis, the four-cluster solution was the best compromise. Descriptive statistics on adherence to the MD before the pandemic and during and after the lockdown of the four identified clusters are shown in Table 2.

The diet of Cluster 1, consisting of about one-third of the sample, dramatically worsened during the pandemic. In fact, in this cluster, the difference between mean adherence to the MD after lockdown and before the pandemic was the largest among the four clusters (DeltaAft_Bef = −6.16). That is why we named it the "Worsened" cluster. Cluster 2, with just over a quarter of the consumers (27%), was characterised by a moderate adherence to

the MD that was constant throughout the periods analysed and intermediate to the other clusters. For these characteristics, the cluster was renamed to "Moderate". Although the "Moderate" consumers started from a lower average adherence score than the "Worsened", from the lockdown period on they showed higher adherence scores than the first cluster. Cluster 3, which accounted for 17% of respondents, comprised the most "Virtuous" consumers. Among the clusters identified, this cluster showed the highest average adherence to MD before the pandemic, which, although decreasing in the two subsequent periods, remained higher than the other three clusters. Finally, Cluster 4 (23% of the sample) contained the "Least careful" to diet. These consumers had the lowest adherence scores even before the pandemic and experienced a further significant worsening during the pandemic, albeit to a lesser extent than the "Worsened" consumers.

**Table 2.** Descriptive statistics of clusters with respect to adherence to the Mediterranean diet.

| | Period | | |
| --- | --- | --- | --- |
| | **Before the Pandemic** | **During the Lockdown** | **After the Lockdown** |
| *Cluster 1 (33%)* | | | |
| Mean | 17.98 | 13.59 | 11.82 |
| St. dev. | 3.41 | 3.44 | 2.88 |
| Median | 18 | 13 | 12 |
| *Cluster 2 (27%)* | | | |
| Mean | 15.19 | 14.05 | 14.69 |
| St. dev. | 2.71 | 1.88 | 2.41 |
| Median | 15 | 14 | 15 |
| *Cluster 3 (17%)* | | | |
| Mean | 20.54 | 19.22 | 19.26 |
| St. dev. | 2.07 | 2.26 | 1.85 |
| Median | 20 | 19 | 19 |
| *Cluster 4 (23%)* | | | |
| Mean | 11.61 | 8.53 | 9.35 |
| St. dev. | 3.72 | 1.89 | 2.97 |
| Median | 11 | 9 | 9 |
| *Overall sample (100%)* | | | |
| Mean | 16.17 | 13.48 | 13.27 |
| St. dev. | 4.38 | 4.24 | 4.22 |
| Median | 16 | 13 | 13 |

Note. The size of each cluster relative to the entire sample is shown in brackets.

After the end of the lockdown, none of the four clusters identified resumed their pre-pandemic adherence to the MD. The four trends identified allow us to respond to RQ1 by describing the changes that have occurred in the eating behaviour during the pandemic.

### 3.2. Diet Changes during the Pandemic

Analysis of pre-pandemic consumption showed that the "Worsened" and "Virtuous" clusters consumed foods with a frequency in line with that recommended for the MD for almost all categories, whereas the "Moderate" and "Least careful" clusters were less adherent. Appendix A shows the results for each cluster related to consumption of the food categories before the pandemic.

### 3.2.1. During the Lockdown

As regards the changes in the diet of the entire sample during the lockdown, in general, we observed an increase in consumption for almost all foods, except for red meat and pork, which recorded a decrease (Appendix B). The analysis of the changes in consumption frequency during the lockdown by individual cluster (Appendix C) showed differences between clusters for all food items, as confirmed by the statistically significant values at 1% in the CHAID analysis.

The "Worsened" consumers generally saw an increase in their consumption of all foods in the survey, except for red meat, pork, and salty snacks, which were found to have decreased. Comparing changes in consumption with other clusters, we found that the "Worsened" cluster was the group containing the highest percentage of those who increased consumption of cereals, vegetables, legumes, potatoes, fruits, dairy products, eggs, fish, and white meat. This group was also the one that decreased the consumption of sweets the most.

The "Moderate" cluster generally increased consumption of all food items except potatoes, cold cuts, red meat, and pork, which remain unchanged. Compared to the sample's eating behaviours, however, this segment showed higher percentages of consumers who kept their intake for all foods unchanged.

In the "Virtuous" cluster, the consumption frequency of all foods was constant overall throughout the lockdown. When looking at the differences with the other clusters, this cluster had the highest percentages of consumers in the "unchanged" consumption category for each food.

In contrast, the "Least careful" consumers generally increased their consumption of all categories. Comparing the behaviour of this cluster with the others, however, it was observed that they were those who decreased the most their consumption of cereals, vegetables, legumes, potatoes, fruit, milk, dairy products, eggs, and fish. The "Least careful" consumers were also those who increased their intake of red meat, pork, salty snacks, and sweets the most. As regards the consumption of cold cuts, this cluster showed two contrasting behaviours, i.e., it had the highest percentages of consumers compared to the other clusters who either increased or decreased their intake of this food.

### 3.2.2. After the Lockdown

Analysing the post-lockdown changes in consumption frequencies for the entire sample versus the lockdown period (Appendix D), we again observed a general trend of increasing consumption, except for red meat and pork (constant) and cold cuts (decreasing). The clusters recorded similar trends to those observed during the lockdown. Again, differences between clusters (Appendix E) were statistically significant at 1% for each food as detected by CHAID analysis.

The "Worsened" cluster generally continued to increase food consumption in all categories except for cold cuts, pork, and salty snacks, which declined. Compared to the other clusters, this was the segment with the highest increase in the intake of most of the foods considered, except for red meat, pork, and salty snacks, which recorded the greatest decreases in consumption. Therefore, when considering the intake of pork compared with the other clusters, the "Worsened" cluster had high percentages of both consumers who increased their consumption and consumers who decreased their intake.

The "Moderate" cluster did not show any change in its diet from the lockdown period, confirming unchanged above-average consumption for all food categories.

The "Virtuous" cluster also did not change food consumption in the post-lockdown period compared to the lockdown. The segment also showed the highest percentages of consumers in the "unchanged" consumption category compared to the other three clusters.

The "Least careful" segment generally increased its consumption of each food, except for cold cuts and pork, whose consumption decreased for a higher percentage of respondents. Looking at the differences with the other clusters, however, this segment had the highest percentage of consumers who decreased consumption of cereals, vegetables, legumes, potatoes, fruit, milk dairy products, eggs, fish, and white meat. This cluster also recorded the highest frequencies of consumers who increased the intake of cold cuts, red meat, salty snacks, and sweets.

### 3.3. Profiling of Clusters

Table 3 shows the statistically significant sociodemographic characteristics differentiating the four groups and Appendix F displays the results of the CHAID analysis outputs

for the significant variables. The "Worsened" cluster comprised predominantly Italians under 45 years of age, with a large household with 1–2 children. These consumers had a medium/high pre-pandemic income, which decreased significantly during the pandemic. The "Moderate" group was mostly made up of US consumers aged over 45 years. They live mainly on their own and their income declined during the lockdown. The "Virtuous" cluster mainly included Italians aged over 45 years with, on average, large households with 1–2 children. The medium–high incomes of this cluster remained unchanged during the lockdown. The "Least careful" cluster consisted mainly of US consumers under 45 years of age who either live alone or in particularly large households with many children. The "Least careful" cluster was the only one that had low incomes prior to the pandemic that increased during the pandemic.

**Table 3.** Cluster profiling by sociodemographic variables.

| Variables | Worsened (%) | Moderate (%) | Virtuous (%) | Least Careful (%) | Overall Sample (%) |
|---|---|---|---|---|---|
| *Country* | | | | | |
| Italy | <u>77.2</u> | 55.8 | <u>86.1</u> | 18.1 | 59.1 |
| USA | 22.8 | <u>44.2</u> | 13.9 | <u>81.9</u> | 40.9 |
| *Age* | | | | | |
| Under 45 | <u>49.3</u> | 47.3 | 26.9 | <u>60.8</u> | 47.6 |
| Over 45 | 50.7 | <u>52.7</u> | <u>73.1</u> | 39.2 | 52.4 |
| *Number of household members* | | | | | |
| 1 | 8.4 | <u>16.5</u> | 9.6 | <u>20.8</u> | 13.7 |
| 2/3 | <u>58.2</u> | 52.7 | <u>61.5</u> | 43.1 | 53.7 |
| 4+ | <u>33.4</u> | 30.8 | 28.9 | <u>36.1</u> | 32.6 |
| *Minors* | | | | | |
| 0 | 8.4 | <u>16.5</u> | 9.6 | <u>20.8</u> | 13.7 |
| 1–2 | <u>73.5</u> | 65.2 | <u>82.7</u> | 52.8 | 68.0 |
| 3+ | 18.1 | <u>18.3</u> | 7.7 | <u>26.4</u> | 18.3 |
| *Income before pandemic* | | | | | |
| Low | 18.3 | 17.1 | 10.6 | <u>25.7</u> | 18.4 |
| Medium/high | <u>81.7</u> | <u>82.9</u> | <u>89.4</u> | 74.3 | 81.6 |
| *Income during lockdown* | | | | | |
| Low | <u>34.7</u> | <u>30.8</u> | 19.7 | 26.0 | 29.1 |
| Medium/high | 65.3 | 69.2 | <u>80.3</u> | <u>74.0</u> | 70.9 |

Note. Only variables found to be significant in the CHAID analysis are reported. Underlined values indicate the presence in the clusters of consumer frequencies above the sample mean frequencies for the given variables.

The statistically significant result in the CHAID analysis of the Country variable indicated that there were differences between Italy and the United States with respect to the MD during the pandemic, thus allowing RQ2 to be answered. Italians were those whose eating behaviours either "Worsened" or were "Virtuous", while the US consumers were either "Moderate" or "Least careful".

Table 4 shows the statistically significant psychographic characteristics differentiating the four groups and Appendix E displays the results of the CHAID analysis outputs for the significant variables. The psychographic variables of the single-item Food Choice Questionnaire [56] showed tendentially common attitudes of the "Worsened" and "Virtuous" clusters with respect to the food attributes. Consumers in these clusters were those who value that the food they eat on a typical day is healthy (Health), helps the mood (Mood), provides them with pleasurable sensations (Sensory), is natural (Natural), is environmental sustainable (Environment), is animal friendly (Animal), and is fairly traded (Social justice). The "Moderate" and "Least careful" clusters were, on the other hand, particularly

interested in the convenience characteristics of foods (Convenience), i.e., products that are practical and quick to cook and consume. The only psychographic feature that was shared by the two clusters was an interest in the positive effect given by food consumption on mood (Mood). This feature was very important for the "Least careful", while it was not for the "Moderate" cluster.

**Table 4.** Cluster profiling by psychographic variables.

| Variables | Worsened (%) | Moderate (%) | Virtuous (%) | Least Careful (%) | Total (%) |
|---|---|---|---|---|---|
| *Health* | | | | | |
| Medium/low | 13.9 | <u>21.0</u> | 12.0 | <u>25.7</u> | 18.2 |
| High | <u>86.1</u> | 79.0 | <u>88.0</u> | 74.3 | 81.8 |
| *Mood* | | | | | |
| Medium/low | 25.3 | <u>40.6</u> | 30.8 | 30.6 | 31.5 |
| High | <u>74.7</u> | 59.4 | <u>69.2</u> | <u>69.4</u> | 68.5 |
| *Convenience* | | | | | |
| Medium/low | <u>44.5</u> | 40.2 | <u>53.9</u> | 28.8 | 41.3 |
| High | 55.5 | <u>59.8</u> | 46.1 | <u>71.2</u> | 58.7 |
| *Sensory* | | | | | |
| Medium/low | 10.6 | <u>15.2</u> | 9.6 | <u>24.3</u> | 14.9 |
| High | <u>89.4</u> | 84.8 | <u>90.4</u> | 75.7 | 85.1 |
| *Natural* | | | | | |
| Medium/low | 19.3 | <u>29.9</u> | 20.2 | <u>38.9</u> | 26.9 |
| High | <u>80.7</u> | 70.1 | <u>79.8</u> | 61.1 | 73.1 |
| *Environment* | | | | | |
| Medium/low | 31.9 | <u>39.3</u> | 27.4 | <u>46.9</u> | 36.6 |
| High | <u>68.1</u> | 60.7 | <u>72.6</u> | 53.1 | 63.4 |
| *Animal* | | | | | |
| Low | 3.7 | <u>8.5</u> | 5.8 | <u>14.9</u> | 8.0 |
| Medium/high | <u>96.3</u> | 91.5 | <u>94.2</u> | 85.1 | 92.0 |
| *Social justice* | | | | | |
| Medium/low | 41.1 | <u>51.5</u> | 39.9 | <u>49.7</u> | 45.7 |
| High | <u>58.9</u> | 48.5 | <u>60.1</u> | 50.3 | 54.3 |

Note. The name of the variables is taken from the scale of Onwezen et al. [56]. Only variables found to be significant in the CHAID analysis are reported. Underlined values indicate the presence in the clusters of consumer frequencies above the sample mean frequencies for the given variables.

Describing each cluster through sociodemographic and psychographic characteristics allowed us to answer RQ3, that is, to outline the profile of those consumers whose adherence to the MD worsened or was unchanged.

## 4. Discussion

This study found an overall increase in food consumption during the pandemic, consistent with other studies, e.g., [14,44]. Although consumption of healthy foods such as fruits and vegetables increased, so did high-calorie foods such as sweets and snacks. Such findings are in line with those of surveys conducted in other countries [63–65]. The consumption trends detected in this study highlight a progressive worsening in adherence to the MD among both Italian and US consumers during the pandemic.

To identify trends in eating behaviours, we segmented our sample of Italian and US consumers into four clusters based on adherence to the MD before the pandemic, and during and after the lockdown. The "Worsened" consumers were those whose adherence to the MD decreased most as a result of increased consumption of nearly all the foods surveyed during the lockdown period. The "Moderate" consumers showed no major changes

in adherence to the MD and maintained consumption of each food constant. The "Virtuous" comprised mostly more diet-conscious consumers, with the highest adherence to MD remaining constant throughout the pandemic phases. Finally, the "Least careful" consumers exhibited the lowest adherence before the pandemic, which declined further during the pandemic.

The clusters identified in this study showed some similarities to segments identified by Grant et al. [38] in an analysis of changes in adherence to the MD during the lockdown in Italy. Their "More eaters" (41% of the sample) have points in common with the "Worsened" cluster in this study. During the lockdown, both clusters, consisting of those under 45 years of age (50 in the "More eaters"), increased consumption of nearly all food categories and exhibited moderate adherence to the MD. In contrast to our study where gender was not significant in profiling, "More eaters" were overwhelmingly women. Common features can also be found between the "Healthy eaters" (27% of the sample) in Grant et al. [38] and our "Virtuous". Both clusters showed high adherence to MD during the lockdown and consisted of consumers in the highest age groups. Again, the group in the study by Grant et al. [38] appeared to be predominantly female. The similarities between our clusters and those of the study by Grant et al. [38] conducted in Italy only were found precisely with our two clusters characterized by the prevalent presence of Italians. This outcome seems to indicate that the predictors that most drove changes in adherence to the MD during the pandemic were nationality and age of consumers, as was detected by the CHAID analysis. In fact, we also observed a worsening of diet only among those under 45 years of age, although Italian consumers, concentrated in the "Worsened" and "Virtuous" clusters, had higher average scores than US consumers before the COVID-19 outbreak. The consumption of healthier foods by older people and poor nutrition by younger people during the lockdown is consistent with findings of studies conducted in the United States [8] and Spain [47].

The different eating behaviours of the clusters seem to depend also on the different food choice motives of Italian and US consumers. Italians' higher initial score also correlated with greater attention to health, the sensory aspect of food, its naturalness, environmental sustainability, and social justice (similar to the consumption motives found by Guiné et al. [66] and Wongprawmas et al. [67]). In contrast, US consumers were more interested in convenience products. The two worsened groups shared only the importance of food's role in mood, and it seems plausible, therefore, that these consumers changed their eating habits to help mood. The desire to choose foods based on factors such as mood was positively correlated with emotional eating. These trends toward emotional eating and comfort food consumption are consistent with findings of many studies on food consumption recorded during the pandemic in both Italy and the United States, e.g., [11–13]. Stress caused by the pandemic may therefore have caused the "Worsened" cluster to eat more in general and the "Least careful" cluster to eat more sweets and salty snacks as our study found. In this sense, in the "Worsened" cluster, there was also an inconsistency between their psychographic characteristics and how they actually act. Together with the "Virtuous" cluster, they declared to be attentive to healthy eating. However, while the "Worsened" cluster increased the consumption of fruits and vegetables, it also consumed high-calorie foods more frequently. The effect of this behaviour was that the "Worsened" cluster recorded the greatest deviation from the principles of the MD. As observed in Marty et al. [45], during the lockdown, consumers modified their food choice motives by increasing the importance of mood and weight control. These modifications may have led opposite changes in the diet. On one hand, stress and boredom caused by the pandemic may explain the increased importance of mood, related to sweet and salty food consumption and decreased quality of diet nutritional quality. On the other hand, the sedentary life of consumer during the lockdown increased the importance of weight control and the consumption of healthy food such as fruit and vegetables.

The other characteristics common to the clusters that saw a worsening in diets can be traced back to households that are large and have children. Tiffin et al. [68] had already

pointed out that the presence of children in households led to worse diets. The "Virtuous" and "Moderate" clusters also had constant incomes, while the other clusters varied, albeit in different directions, thus certainly affecting consumption during the lockdown. Indeed, the link between unhealthy diets and low incomes is widely acknowledged in the literature [69].

In general, however, we observed a downward trend in adherence to the MD in the entire sample, with very few exceptions, as no cluster showed an improvement in its diet. The literature reports mixed results on this topic, as outlined in the introduction, due mainly to the methodologies used in research [34]. While studies that have used MEDAS show an overall improvement [9,35,36], others have shown a worsening [46,47]. Therefore, our results are in line with studies that did not use MEDAS and that have reported a deviation from a healthy diet. In addition, this is one of the first papers that also analyses what happened after the lockdown, having collected the data in September 2020, when the general perception was that the emergency was over. The other two papers on this topic [48,49] are in line with our findings, showing a worsening trend in diet.

## 5. Limitation and Suggestion for Future Research

Our aim was to investigate changes in adherence to the MD during and after the lockdown, while also analysing the causes of any worsening or improvement. In addition to sociodemographic characteristics, we included psychographic characteristics using the single-item food choice questionnaire. However, one limitation of our work is that we did not survey BMI and weight changes during the lockdown phases, which would have provided us with a clearer picture of the impact of these negative changes in diet. Future research on this topic could investigate this aspect as well, perhaps also adding questions related to physical activity, in order to understand whether the phenomenon of emotional eating and a general increase in consumption have actually had negative impacts on body weight in all consumers.

Although, as mentioned, our study is one of the first to investigate the situation of diet in the post-lockdown period, further studies will be needed once the COVID-19 pandemic is officially under control. This will allow us to confirm our findings and investigate further what means and instruments might be available to fix poor diets. Therefore, starting from the results of our work, it would be interesting to study the post-COVID-19 eating behaviours of consumers under the age of 45, with children, thus gaining greater knowledge of the consumer segment that saw a worsening in adherence to the MD. In addition, we focused on Italy and the United States and found that, although the starting point was different, eating behaviours during the lockdown were not dependent on the country of residence. It would be useful to carry out the same survey in other countries or states that experienced a total lockdown to see whether or not these findings hold true.

## 6. Conclusions

Having demonstrated that at the end of the pandemic, the turn for the worse in diets will be confirmed, it is necessary for policy-makers to take action to curb the problem in time. The phenomenon of emotional eating did not seem to come to an end with the lifting of the lockdown. Increased consumption of junk food and other unhealthy foods, coupled with scarce physical activity due to the restrictions, can only lead to weight gain, resulting in negative health impacts [70]. It would therefore be desirable to take action, above all among those we have seen to be most exposed to a worsening of diet, that is, people under 45 years of age, with large families and children, through information campaigns encouraging healthy diets. Among other things, we showed that the cluster that drastically worsened ("Worsened") is composed of consumers who actually claim to be health conscious and attentive to other factors that would allow for a balanced diet. Therefore, it would be important to recall these aspects, so as to stir these respondents up and make them understand the need to pay attention to what they eat and to modulate their food consumption as much as possible. In conclusion, the adoption of the MD, a diet with beneficial proprieties for the immune system, can be a possible strategy for attenuating the

severity of COVID-19 infection of affected population. Policy-makers should combine such therapeutic aspect of MD with campaigns aimed to increase social awareness for healthy eating and preventing obesity.

**Author Contributions:** Conceptualization, G.S.; methodology, F.G., T.F. and G.S.; software, C.C. and G.S.; validation, L.C. and T.F.; formal analysis, F.G. and T.F.; data curation, C.C. and G.S.; writing—original draft preparation, F.G.; writing—review and editing, F.G. and T.F.; supervision, G.S. All authors have read and agreed to the published version of the manuscript.

**Funding:** This research received no external funding.

**Institutional Review Board Statement:** Not applicable.

**Informed Consent Statement:** Informed consent was obtained from all participants involved in the study.

**Data Availability Statement:** The data presented in this study are available on request from the corresponding author.

**Conflicts of Interest:** The authors declare no conflict of interest.

## Appendix A. Food Consumption before the Pandemic

**Table A1.** Frequency of consumption of each food in each cluster before the pandemic.

| Food | Worsened (%) | Moderate (%) | Virtuous (%) | Least Careful (%) | Overall Sample (%) |
|---|---|---|---|---|---|
| *Cereals* | | | | | |
| Never | 0.3 | 1.5 | 0.0 | 6.3 | 1.9 |
| Less than once a week | 6.4 | 10.1 | 4.3 | 21.2 | 10.5 |
| Sometimes in a week | 29.5 | 47.9 | 30.3 | 41.3 | 37.3 |
| Once a day | 41.3 | 32.9 | 43.3 | 21.5 | 34.8 |
| More than once a day | 22.5 | 7.6 | 22.1 | 9.7 | 15.5 |
| *Vegetables* | | | | | |
| Never | 0.7 | 0.6 | 0.0 | 3.1 | 1.2 |
| Less than once a week | 3.5 | 4.9 | 0.0 | 9.7 | 4.7 |
| Sometimes in a week | 27.0 | 39.3 | 14.4 | 36.1 | 30.3 |
| Once a day | 37.9 | 37.5 | 49.0 | 32.7 | 38.4 |
| More than once a day | 30.9 | 17.7 | 36.6 | 18.4 | 25.4 |
| *Legumes* | | | | | |
| Never | 5.5 | 9.5 | 1.4 | 27.4 | 11.0 |
| Less than once a week | 20.0 | 34.1 | 11.5 | 24.3 | 23.4 |
| Sometimes in a week | 61.4 | 48.5 | 80.3 | 26.8 | 53.0 |
| Once a day | 11.1 | 6.1 | 5.8 | 15.6 | 9.9 |
| More than once a day | 2.0 | 1.8 | 1.0 | 5.9 | 2.7 |
| *Potatoes* | | | | | |
| Never | 0.5 | 0.9 | 0.0 | 5.9 | 1.8 |
| Less than once a week | 18.1 | 32.9 | 15.4 | 29.2 | 24.2 |
| Sometimes in a week | 71.0 | 58.0 | 82.7 | 38.5 | 61.9 |
| Once a day | 8.4 | 7.9 | 1.9 | 18.1 | 9.4 |
| More than once a day | 2.0 | 0.3 | 0.0 | 8.3 | 2.7 |

**Table A1.** *Cont.*

| Food | Worsened (%) | Moderate (%) | Virtuous (%) | Least Careful (%) | Overall Sample (%) |
|---|---|---|---|---|---|
| *Fruit* | | | | | |
| Never | 0.0 | 0.6 | 0.0 | 3.5 | 1.0 |
| Less than once a week | 4.5 | 8.2 | 1.0 | 14.2 | 7.1 |
| Sometimes in a week | 19.1 | 28.7 | 7.2 | 38.9 | 24.3 |
| Once a day | 35.1 | 39.9 | 35.6 | 27.4 | 34.7 |
| More than once a day | 41.3 | 22.6 | 56.2 | 16.0 | 32.9 |
| *Milk* | | | | | |
| Never | 5.7 | 8.5 | 2.9 | 16.7 | 8.6 |
| Less than once a week | 11.4 | 12.2 | 10.1 | 10.1 | 11.1 |
| Sometimes in a week | 18.8 | 34.5 | 16.3 | 27.8 | 24.7 |
| Once a day | 51.2 | 32.3 | 63.0 | 28.1 | 42.7 |
| More than once a day | 12.9 | 12.5 | 7.7 | 17.3 | 12.9 |
| *Dairy products* | | | | | |
| Never | 0.3 | 0.9 | 0.0 | 8.3 | 2.3 |
| Less than once a week | 8.4 | 10.7 | 2.9 | 14.6 | 9.5 |
| Sometimes in a week | 53.0 | 63.1 | 60.6 | 42.7 | 54.6 |
| Once a day | 32.4 | 19.5 | 30.8 | 24.0 | 26.7 |
| More than once a day | 5.9 | 5.8 | 5.7 | 10.4 | 6.9 |
| *Eggs* | | | | | |
| Never | 0.7 | 1.2 | 0.5 | 6.6 | 2.2 |
| Less than once a week | 19.6 | 28.4 | 19.7 | 18.0 | 21.6 |
| Sometimes in a week | 64.4 | 53.4 | 76.9 | 42.4 | 58.4 |
| Once a day | 12.6 | 14.9 | 1.9 | 22.9 | 13.8 |
| More than once a day | 2.7 | 2.1 | 1.0 | 10.1 | 4.0 |
| *Fish* | | | | | |
| Never | 2.2 | 5.2 | 0.0 | 17.0 | 6.1 |
| Less than once a week | 29.5 | 39.3 | 26.4 | 33.0 | 32.4 |
| Sometimes in a week | 59.4 | 45.8 | 73.6 | 29.2 | 51.1 |
| Once a day | 6.4 | 8.2 | 0.0 | 13.9 | 7.6 |
| More than once a day | 2.5 | 1.5 | 0.0 | 6.9 | 2.8 |
| *Cold cuts* | | | | | |
| Never | 3.2 | 7.0 | 1.4 | 22.9 | 8.6 |
| Less than once a week | 24.5 | 26.8 | 25.6 | 22.9 | 24.9 |
| Sometimes in a week | 60.1 | 55.5 | 64.4 | 29.9 | 52.5 |
| Once a day | 9.7 | 8.9 | 7.2 | 19.8 | 11.4 |
| More than once a day | 2.5 | 1.8 | 1.4 | 4.5 | 2.6 |
| *White meat* | | | | | |
| Never | 0.7 | 1.2 | 0.0 | 12.5 | 3.5 |
| Less than once a week | 9.4 | 26.6 | 10.1 | 12.8 | 14.9 |
| Sometimes in a week | 77.0 | 57.6 | 83.2 | 46.2 | 65.6 |
| Once a day | 11.4 | 13.4 | 6.2 | 19.1 | 12.9 |
| More than once a day | 1.5 | 1.2 | 0.5 | 9.4 | 3.1 |

**Table A1.** *Cont.*

| Food | Worsened (%) | Moderate (%) | Virtuous (%) | Least Careful (%) | Overall Sample (%) |
|---|---|---|---|---|---|
| *Red meat* | | | | | |
| Never | 2.5 | 4.6 | 1.4 | 18.8 | 6.7 |
| Less than once a week | 29.7 | 34.8 | 37.0 | 19.4 | 29.9 |
| Sometimes in a week | 62.1 | 53.67 | 61.1 | 40.3 | 54.56 |
| Once a day | 4.5 | 5.8 | 0.5 | 15.6 | 6.8 |
| More than once a day | 1.2 | 1.2 | 0.0 | 5.9 | 2.1 |
| *Pork* | | | | | |
| Never | 8.2 | 11.9 | 1.9 | 27.1 | 12.5 |
| Less than once a week | 37.4 | 40.2 | 51.0 | 27.4 | 38.1 |
| Sometimes in a week | 50.0 | 43.6 | 47.1 | 28.5 | 42.8 |
| Once a day | 4.2 | 3.7 | 0.0 | 14.2 | 5.7 |
| More than once a day | 0.2 | 0.6 | 0.0 | 2.8 | 0.9 |
| *Salty snacks* | | | | | |
| Never | 6.4 | 7.3 | 3.8 | 7.6 | 6.5 |
| Less than once a week | 31.2 | 28.0 | 40.4 | 22.9 | 30.0 |
| Sometimes in a week | 43.1 | 47.6 | 48.1 | 36.1 | 43.5 |
| Once a day | 16.8 | 13.4 | 7.7 | 24.7 | 16.2 |
| More than once a day | 2.5 | 3.7 | 0.0 | 8.7 | 3.8 |
| *Sweets* | | | | | |
| Never | 0.7 | 0.9 | 0.0 | 9.0 | 2.6 |
| Less than once a week | 29.0 | 22.9 | 26.0 | 19.1 | 24.5 |
| Sometimes in a week | 44.8 | 50.6 | 57.7 | 37.2 | 46.7 |
| Once a day | 20.10 | 20.1 | 14.4 | 21.9 | 19.65 |
| More than once a day | 5.4 | 5.5 | 1.9 | 12.8 | 6.6 |

Note. Underlined consumption frequencies in the first column indicate consumption that most closely adheres to the MD according to the Cavaliere et al. [54] scale.

## Appendix B. Food Consumption Changes of the Entire Sample during the Pandemic

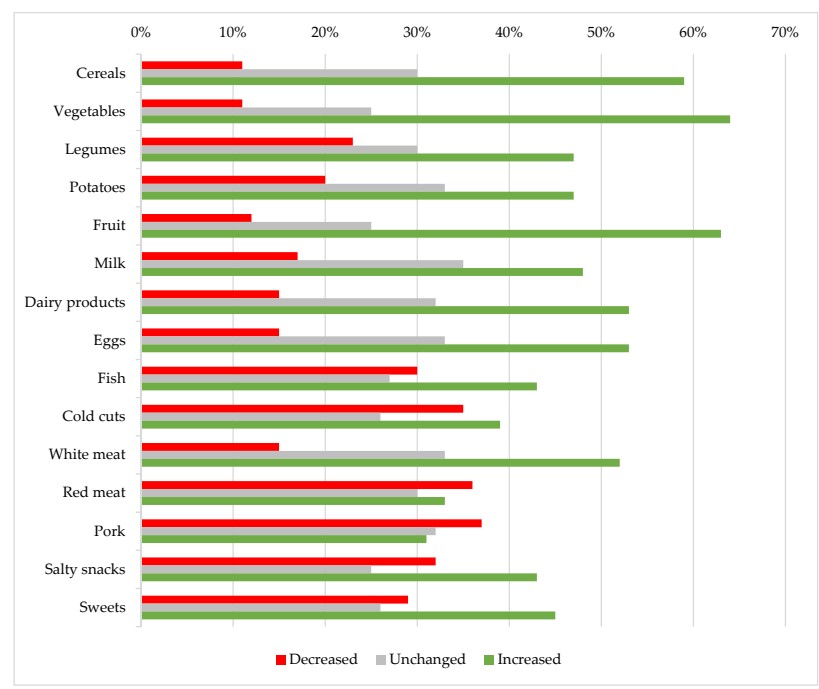

**Figure A1.** Changes in consumption frequencies of the entire sample during the lockdown compared to before the pandemic period.

## Appendix C. Changes in Consumption Frequencies during the Lockdown of the Clusters

**Table A2.** Changes in consumption frequencies of each food item during the lockdown compared with the pre-pandemic period.

| Food | Worsened (%) | Moderate (%) | Virtuous (%) | Least Careful (%) | Overall Sample (%) |
|---|---|---|---|---|---|
| *Cereals* | | | | | |
| Decreased | 10.9 | 9.2 | 6.7 | <u>17.0</u> | 11.2 |
| Unchanged | 18.1 | 34.1 | <u>55.8</u> | 21.9 | 29.6 |
| Increased | <u>71.0</u> | 56.7 | 37.5 | 61.1 | 59.2 |
| *Vegetables* | | | | | |
| Decreased | 8.7 | 10.1 | 5.8 | <u>19.1</u> | 11.0 |
| Unchanged | 11.4 | 31.7 | <u>50.5</u> | 17.7 | 24.9 |
| Increased | <u>79.9</u> | 58.2 | 43.7 | 63.2 | 64.1 |
| *Legumes* | | | | | |
| Decreased | 20.3 | 21.0 | 8.6 | <u>38.5</u> | 22.8 |
| Unchanged | 17.1 | 37.2 | <u>61.1</u> | 19.1 | 30.4 |
| Increased | <u>62.6</u> | 41.8 | 30.3 | 42.4 | 46.8 |
| *Potatoes* | | | | | |
| Decreased | 18.3 | 19.8 | 11.5 | <u>26.7</u> | 19.5 |
| Unchanged | 21.3 | 40.9 | <u>58.7</u> | 21.9 | 33.0 |
| Increased | <u>60.4</u> | 39.3 | 29.8 | 51.4 | 47.5 |
| *Fruit* | | | | | |
| Decreased | 10.1 | 11.6 | 6.7 | <u>18.4</u> | 11.9 |
| Unchanged | 13.9 | 28.7 | <u>51.0</u> | 17.4 | 24.9 |
| Increased | <u>76.0</u> | 59.7 | 42.3 | 64.2 | 63.2 |
| *Milk* | | | | | |
| Decreased | 18.8 | 15.9 | 8.6 | <u>20.5</u> | 16.7 |
| Unchanged | 22.8 | 39.0 | <u>65.9</u> | 27.1 | 35.4 |
| Increased | <u>58.4</u> | 45.1 | 25.5 | 52.4 | 47.9 |
| *Dairy products* | | | | | |
| Decreased | 17.1 | 14.0 | 8.2 | <u>17.4</u> | 14.8 |
| Unchanged | 19.0 | 37.2 | <u>59.6</u> | 25.0 | 32.2 |
| Increased | <u>63.9</u> | 48.8 | 32.2 | 57.6 | 53.0 |
| *Eggs* | | | | | |
| Decreased | 15.1 | 13.1 | 10.1 | <u>19.8</u> | 14.8 |
| Unchanged | 20.8 | 37.8 | <u>66.8</u> | 18.7 | 32.7 |
| Increased | <u>64.1</u> | 49.1 | 23.1 | 61.5 | 52.5 |
| *Fish* | | | | | |
| Decreased | 30.2 | 29.0 | 24.0 | <u>36.1</u> | 30.2 |
| Unchanged | 15.3 | 32.9 | <u>50.5</u> | 20.1 | 27.1 |
| Increased | <u>54.5</u> | 38.1 | 25.5 | 43.8 | 42.7 |
| *Cold cuts* | | | | | |
| Decreased | 38.1 | 32.6 | 28.9 | <u>38.5</u> | 35.2 |
| Unchanged | 15.6 | 33.9 | <u>51.9</u> | 13.9 | 26.2 |
| Increased | 46.3 | 33.5 | 19.2 | <u>47.6</u> | 38.6 |
| *White meat* | | | | | |
| Decreased | 15.8 | 12.5 | 10.1 | <u>21.9</u> | 15.4 |
| Unchanged | 18.3 | 41.2 | <u>64.4</u> | 20.8 | 32.8 |
| Increased | <u>65.9</u> | 46.3 | 25.5 | 57.3 | 51.8 |
| *Red meat* | | | | | |
| Decreased | <u>43.3</u> | 31.7 | 27.9 | 37.5 | 36.2 |
| Unchanged | 19.1 | 36.3 | <u>59.1</u> | 18.7 | 30.4 |
| Increased | 37.6 | 32.0 | 13.0 | <u>43.8</u> | 33.4 |
| *Pork* | | | | | |
| Decreased | <u>43.1</u> | 34.8 | 27.4 | 37.5 | 36.9 |
| Unchanged | 21.3 | 39.3 | <u>59.6</u> | 20.1 | 32.3 |
| Increased | 35.6 | 25.9 | 13.0 | <u>42.4</u> | 30.8 |

**Table A2.** *Cont.*

| Food | Worsened (%) | Moderate (%) | Virtuous (%) | Least Careful (%) | Overall Sample (%) |
|---|---|---|---|---|---|
| *Salty snacks* | | | | | |
| Decreased | <u>42.6</u> | 25.6 | 30.3 | 26.8 | 32.2 |
| Unchanged | 13.6 | 33.5 | <u>46.1</u> | 14.9 | 24.8 |
| Increased | 43.8 | 40.9 | 23.6 | <u>58.3</u> | 43.0 |
| *Sweets* | | | | | |
| Decreased | <u>35.4</u> | 28.3 | 17.8 | 29.2 | 29.1 |
| Unchanged | 15.6 | 30.2 | <u>54.8</u> | 16.3 | 26.3 |
| Increased | 49.0 | 41.5 | 27.4 | <u>54.5</u> | 44.6 |

Note. For easy reading, the underlined values indicate the frequencies of consumers in one cluster that are greater than in the other three clusters.

## Appendix D. Food Consumption Changes of the Entire Sample in the Post-Lockdown Period

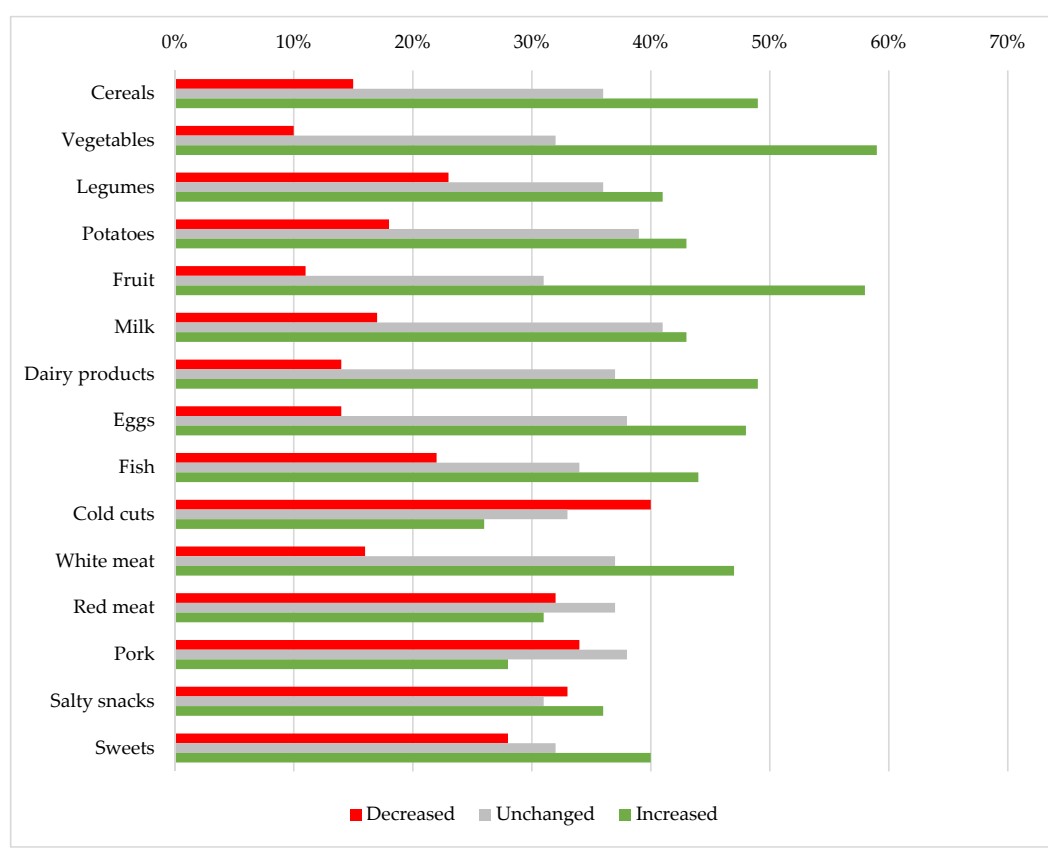

**Figure A2.** Changes in consumption frequencies of the entire sample after the lockdown compared to during the lockdown period.

## Appendix E. Changes in Consumption Frequencies in the Post-Lockdown Period of the Clusters

**Table A3.** Changes in consumption frequency of each food in the post-lockdown period versus the lockdown period.

| Food | Worsened (%) | Moderate (%) | Virtuous (%) | Least Careful (%) | Overall Sample (%) |
|---|---|---|---|---|---|
| *Cereals* | | | | | |
| Decreased | 16.3 | 14.0 | 5.8 | <u>20.5</u> | 14.9 |
| Unchanged | 21.3 | 42.7 | <u>66.3</u> | 28.1 | 36.2 |
| Increased | <u>62.4</u> | 43.3 | 27.9 | 51.4 | 48.9 |

**Table A3.** *Cont.*

| Food | Worsened (%) | Moderate (%) | Virtuous (%) | Least Careful (%) | Overall Sample (%) |
|---|---|---|---|---|---|
| *Vegetables* | | | | | |
| Decreased | 9.9 | 7.3 | 3.9 | <u>15.6</u> | 9.5 |
| Unchanged | 13.9 | 39.9 | <u>63.9</u> | 24.0 | 31.7 |
| Increased | <u>76.2</u> | 52.8 | 32.2 | 60.4 | 58.8 |
| *Legumes* | | | | | |
| Decreased | 21.5 | 20.7 | 7.7 | <u>37.8</u> | 22.8 |
| Unchanged | 19.8 | 47.6 | <u>69.2</u> | 23.3 | 36.4 |
| Increased | <u>58.7</u> | 31.7 | 23.1 | 38.9 | 40.8 |
| *Potatoes* | | | | | |
| Decreased | 21.5 | 16.8 | 6.2 | <u>23.9</u> | 18.2 |
| Unchanged | 23.5 | 47.5 | <u>73.1</u> | 27.1 | 39.2 |
| Increased | <u>55.0</u> | 35.7 | 20.7 | 49.0 | 42.6 |
| *Fruit* | | | | | |
| Decreased | 11.6 | 9.5 | 3.4 | <u>16.7</u> | 10.8 |
| Unchanged | 15.4 | 37.5 | <u>60.1</u> | 24.3 | 31.0 |
| Increased | <u>73.0</u> | 53.0 | 36.5 | 59.0 | 58.2 |
| *Milk* | | | | | |
| Decreased | 19.3 | 14.3 | 6.2 | <u>22.9</u> | 16.6 |
| Unchanged | 27.0 | 46.1 | <u>75.5</u> | 28.1 | 40.6 |
| Increased | <u>53.7</u> | 39.6 | 18.3 | 49.0 | 42.8 |
| *Dairy products* | | | | | |
| Decreased | 19.1 | 10.1 | 5.3 | <u>19.4</u> | 14.4 |
| Unchanged | 18.5 | 46.9 | <u>71.1</u> | 25.0 | 36.6 |
| Increased | <u>62.4</u> | 43.0 | 23.6 | 55.6 | 49.0 |
| *Eggs* | | | | | |
| Decreased | 17.3 | 12.5 | 4.8 | <u>18.7</u> | 14.3 |
| Unchanged | 20.3 | 47.0 | <u>78.9</u> | 21.9 | 37.7 |
| Increased | <u>62.4</u> | 40.5 | 16.3 | 59.4 | 48.0 |
| *Fish* | | | | | |
| Decreased | 25.5 | 15.0 | 13.0 | <u>31.3</u> | 21.9 |
| Unchanged | 14.8 | 46.0 | <u>66.8</u> | 24.6 | 34.3 |
| Increased | <u>59.7</u> | 39.0 | 20.2 | 44.1 | 43.8 |
| *Cold cuts* | | | | | |
| Decreased | <u>49.0</u> | 38.1 | 26.9 | 39.9 | 40.2 |
| Unchanged | 18.8 | 40.9 | <u>64.0</u> | 22.9 | 33.3 |
| Increased | 32.2 | 21.0 | 9.1 | <u>37.2</u> | 26.5 |
| *White meat* | | | | | |
| Decreased | 16.6 | 14.9 | 7.7 | <u>23.6</u> | 16.3 |
| Unchanged | 18.8 | 47.9 | <u>71.1</u> | 25.0 | 36.9 |
| Increased | <u>64.6</u> | 37.2 | 21.2 | 51.4 | 46.8 |
| *Red meat* | | | | | |
| Decreased | <u>40.6</u> | 25.9 | 21.6 | 35.8 | 32.3 |
| Unchanged | 18.6 | 48.8 | <u>71.6</u> | 22.9 | 36.7 |
| Increased | 40.8 | 25.3 | 6.8 | <u>41.3</u> | 31.0 |
| *Pork* | | | | | |
| Decreased | <u>44.1</u> | 28.1 | 21.1 | 37.5 | 34.4 |
| Unchanged | 17.8 | 51.2 | <u>70.2</u> | 26.0 | 37.5 |
| Increased | <u>38.1</u> | 20.7 | 8.7 | 36.5 | 28.1 |
| *Salty snacks* | | | | | |
| Decreased | <u>44.3</u> | 25.6 | 26.4 | 28.5 | 32.6 |
| Unchanged | 13.1 | 43.0 | <u>60.1</u> | 22.9 | 31.3 |
| Increased | 42.6 | 31.4 | 13.5 | <u>48.6</u> | 36.1 |
| *Sweets* | | | | | |
| Decreased | <u>34.7</u> | 25.9 | 18.8 | 27.8 | 28.0 |
| Unchanged | 15.3 | 43.9 | <u>63.4</u> | 20.5 | 32.3 |
| Increased | 50.0 | 30.2 | 17.8 | <u>51.7</u> | 39.7 |

Note. For easy reading, the underlined values indicate the frequencies of consumers in one cluster that are greater than in the other three clusters.

## Appendix F. CHAID Analysis Outputs

**Table A4.** Variables found to be significant in the CHAID analysis with their respective significance levels.

| Variables | LR Chi-Squared | df | Prob. |
|---|---|---|---|
| *Sociodemographic variables* | | | |
| Country | 337.45 | 3 | 0.00 |
| Age | 57.82 | 3 | 0.00 |
| Number of household members | 36.04 | 6 | 0.00 |
| Minors | 67.92 | 6 | 0.00 |
| Income before pandemic | 19.42 | 3 | 0.00 |
| Income during lockdown | 17.22 | 3 | 0.00 |
| *Psychographic variables* | | | |
| Health | 23.00 | 3 | 0.00 |
| Mood | 19.71 | 3 | 0.00 |
| Convenience | 34.47 | 3 | 0.00 |
| Sensory | 28.91 | 3 | 0.00 |
| Natural | 38.70 | 3 | 0.00 |
| Environment | 25.73 | 3 | 0.00 |
| Animal | 29.36 | 3 | 0.00 |
| Social justice | 12.60 | 3 | 0.02 |

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
