# Peer review of "Adherence to the Mediterranean Diet and COVID-19: A Segmentation Analysis of Italian and US Consumers"

_sustainability, doi:10.3390/su14073823_

Round 1

Reviewer 1 Report

Abbreviate Figure 1 (Fig 1.) on page 7 line 256.

Review the values in the tables again. A total value of 100 was not obtained in some categories. For example, the total of dairy products in the worsened group is 99.9 (Table A1).

The research findings were determined as expected. But recommendations need to be developed.

Check the reference section based on the journal rule. For example, there is a difference between references 47 and 52.

Author Response

Please find our replies in the attached file. Thank you

Reviewer 2 Report

Dear Authors, 
The text presented for review is one of many addressing the issue of dietary changes during the Covid-19 pandemic. Its undoubted advantage is the use of a reference to the Mediterranean diet, considered beneficial from a nutritional point of view, and the fact that the study examined the use of this diet both before, during, and after the pandemic. Thus, the very idea of the study is important from a cognitive point of view. 
However, the presentation and interpretation of the results raises some doubts.  All of them will be presented below with reference to particular parts of the text.

1, 2 (Introduction, Adherence to the Mediterranean diet during the lockdown).
The chapter Introductions does not raise objections, whereas the chapter Adherence to the Mediterranean diet during the lockdown is not essential. Some of the information contained therein (about the Mediterranean diet) can be placed in a shortened form in the introduction, some can be moved to the methodology and some to the discussion).  Finally, the Introduction chapter should end with a precise definition of the aim of the study and possibly presentation of the research questions. 

3. study and sample. 
This chapter lacks clear information about maintaining ethical standards while conducting the study. It is also absent at the end of the text. This information is essential and needs to be supplemented. 
In Table 1, as well as in other tables, it is unnecessary to add "%" next to the values - it is enough to signal it at the table title. 
In subsection 3.2, a more extensive explanation of the scale used to collect the psychometric data is needed. This will enable understanding of the information presented in Table 6. 
Perhaps presenting the idea of the study in a diagram would be advisable?
Doubts are raised by the reference to the literature used in line 187, whether it should not read " ... using the scale developed by Onwezen et al. [57] regarding the importance of ...", analogous form of references appears in the text repeatedly. 
4 Results 
Is the data in Figure 1 not a repetition of the information in Table 2. If so, what is the point of adding this chart? 
Tables 3 and 4 do not contain data crucial to the idea of the article, so consider including them as an appendix. 
The data that are key to the research questions formulated by the authors are in Table 6 . However, their discussion is much less extensive (subsection 4.3) than that of dietary changes (4.2.). Perhaps more attention should be paid to the issues raised in this chapter, although this is only a suggestion.  In contrast, recalling the methodology at the beginning of the subsection does not seem necessary. The terms used in the table (e.g. animal, mood, natural) also need clarification. 
Discussion and conclusions
This chapter raises some questions. Its first part basically repeats the descriptions of the individual clusters presented earlier. This is perfectly reasonable, but rather not in such an extensive form. In the following paragraphs the authors compare the obtained results with the results of previous studies, although the number of references (considering the large number of publications on this issue) is small. The discussion ends at line 469, and then the authors consider at length other issues of nutritional and health character. 
I suggest introducing a standard division of the text and separating the Discussion, Limitations and Futher research and Conclusions chapters. This will make the test more organized and prompt the author to clarify the conclusions (synthetically and correctly stated in the abstract). 
Author Contributions - the wording about the equal contribution of the authors to the preparation of the text raises the question on what basis the order of authors was established (it is not an alphabetical order).  

Author Response

(The authors gave the same response as above.)

Reviewer 3 Report

Here Francesca Gerini and colleagues conducted a questionnaire to a representative sample of Italians and New Yorkers studying adherence to the Mediterranean Diet and COVID-19. Raising the awareness of citizens on the issue of correct eating habits and developing adequate food policies to stem the trend towards unhealthy diets is highlighted.

This work is informative that worth publishing within Sustainability readership and community. However, there are some minor concerns to be addressed before acceptance for publication.

Line 28, typo, which is a cause for concern…

Since the authors are studying the COVID-19’s impact on eating habit, it is also necessary to summarize Mediterranean Diet’s attenuation and protection on COVID-19.

Please refer to:

Inverse Association Between the Mediterranean Diet and COVID-19 Risk in Lebanon: A Case-Control Study

Adherence to the Mediterranean Diet during COVID-19 national lockdowns: a systematic review

Author Response

(The authors gave the same response as above.)

Round 2

Reviewer 2 Report

Dear Authors,
I am satisfied with the changes made in the text. I would only suggest to change the order of chapters 5 and 6 (it seems to me that the text should end with conclusions).   
Congratulations
